# Exploring the concordance of recommendations across guidelines on chest imaging for the diagnosis and management of COVID-19: A proposed methodological approach based on a case study

**Sally Yaacoub**[1], **Fatimah Chamseddine**[1], **Farah Jaber**[2], **Ivana Blazic**[3], **Guy Frija**[4], **Elie A. Akl**[2,5]*

**1** Clinical Research Institute, American University of Beirut, Beirut, Lebanon, **2** Department of Internal Medicine, American University of Beirut, Beirut, Lebanon, **3** Clinical Hospital Centre Zemun, Belgrade, Serbia, **4** Université de Paris, Paris, France, **5** Department of Health Research Methods, Evidence & Impact, McMaster University, Ontario, Canada

* ea32@aub.edu.lb

**Data Availability Statement:** All relevant data are within the paper and its Supporting Information files.

## Abstract

### Objective

To describe a methodological approach to explore the concordance of recommendations across guidelines and its application to the case of the WHO recommendations on chest imaging for the diagnosis and management of COVID-19.

### Study design and setting

We followed a methodological approach applied to a case study that included: defining the 'reference guideline' (i.e., the WHO guidance) and the 'reference recommendations'; searching for 'related guidelines' and identifying 'related recommendations'; constructing the PICO for the recommendations; assessing the matching of the PICO of each related recommendation to the PICO corresponding reference recommendation; and assessing the concordance between the PICO-matching recommendations.

### Results

We identified a total of 89 related recommendations from 22 related guidelines. Out of the 89 related recommendations, 43 partly matched and 1 entirely matched one of the reference recommendations, and out of these, 8 were concordant with one of the reference recommendations. When considering the seven reference recommendations, they had a median of 12 related recommendations (range 3–17), a median of 7 PICO-matching recommendations (range 0–13), and a median of 1 concordant recommendation (range 0–4).

**Funding:** The authors received no specific funding for this work.

**Competing interests:** The authors have declared that no competing interests exist.

## Conclusion

Following a detailed methodological approach, we were able to explore the concordance between our reference recommendations and related recommendations from other guidelines. A relatively low percentage of recommendations was concordant.

## Introduction

As the coronavirus disease-2019 (COVID-19) pandemic became a major worldwide threat, guideline developers reacted by developing rapid guidelines addressing a range of relevant topics. The evidence synthesis and guideline communities launched initiatives to respond to the epidemic and support health decision makers in managing the pandemic [1, 2].

As part of the response, many guidelines organizations developed recommendations on the same topics. Consequently, decision makers had to deal with discordant recommendations, i.e., recommendations addressing the same question but providing inconsistent guidance. A recent study comparing the recommendations from guidelines on the intensive care treatment of COVID-19 patients reported conflicting recommendations on the use of antibiotic prophylaxis and a striking discrepancy in the recommendations on the use of oxygen therapy [3]. Another scoping review on guidelines for COVID-19 management in children reported the presence of heterogeneous recommendations across guidelines [4].

Chest imaging for the diagnosis and management of COVID-19 is a case in point. This group of authors was in charge of coordinating the development of the World Health Organization (WHO) guidance on the use of chest imaging for the diagnosis and management of COVID-19 [5]. We followed standard WHO process which includes an early exercise to prioritize questions to be addressed by the recommendations.

As part of the preparatory work for the WHO guideline development, we searched for related guidance. The main purpose was to ensure the WHO project was not duplicating work done by other guideline groups, i.e., not addressing the same questions. Another purpose was to understand the factors that influenced the development of the 'related recommendations' (i.e., the recommendations from the related guidance). This would help us ensure all relevant issues are addressed by the WHO guideline development group as it develops and finalizes its own recommendations. Following the development of the WHO guidance, we went back to compare our recommendations to the previously identified related recommendations. Completing this comparison required the development of a methodological approach.

The objective of this study is to describe a methodological approach to explore the concordance of recommendations across guidelines and its application to the case of the WHO recommendations on chest imaging for the diagnosis and management of COVID-19.

## Methods

We drafted the methodological approach based on a review of the literature [6, 7], discussion amongst the authors and iterative revisions when applying it to the WHO recommendations on chest imaging for the diagnosis and management of COVID-19.

### Methodological approach

The approach is depicted in Fig 1 and described in detail in subsequent sections. Briefly, after defining the 'reference guideline' (i.e., the WHO guidance) and the 'reference

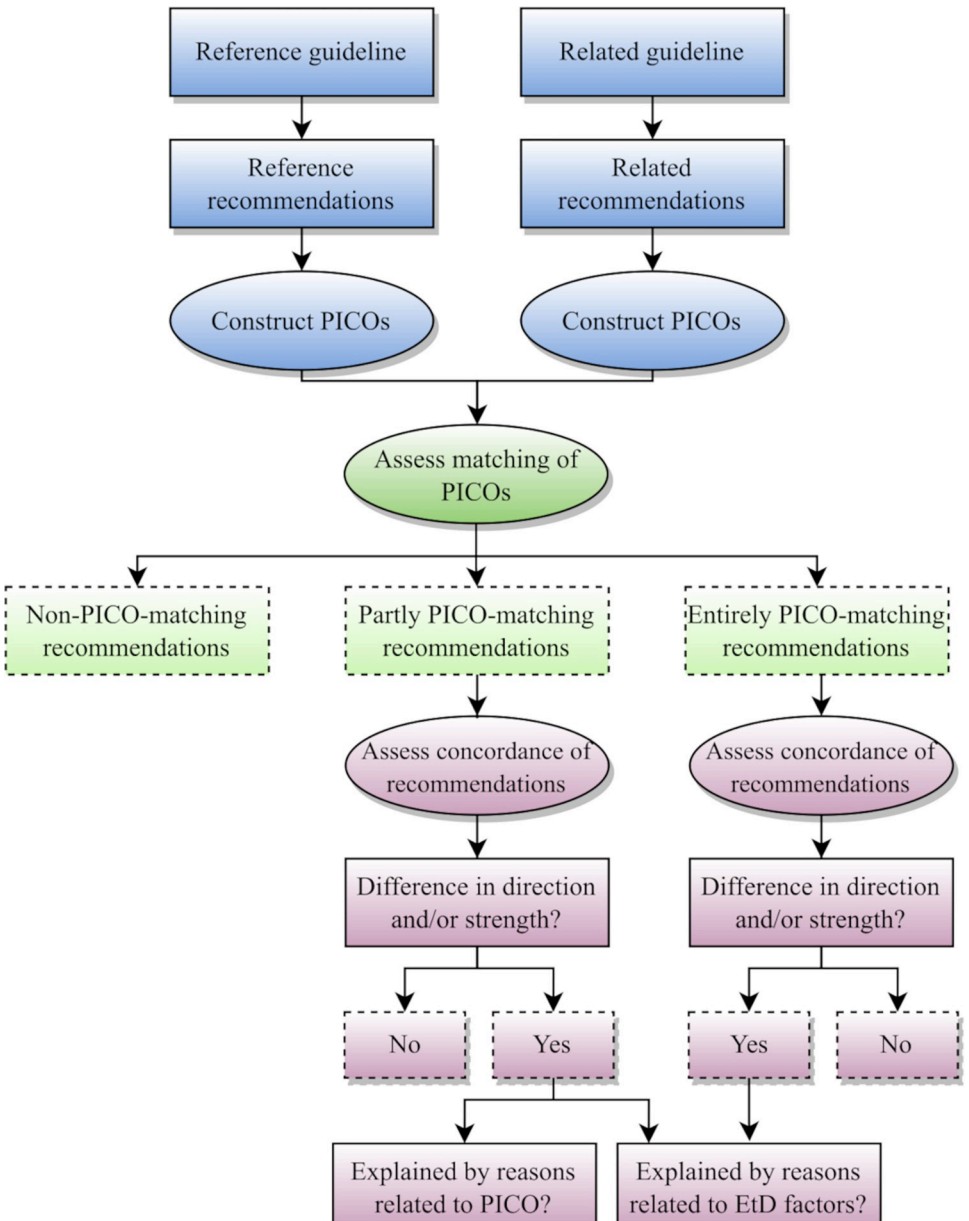

**Fig 1. Methodological approach to explore concordance between recommendations.**

recommendations', we searched for 'related guidelines' and identified 'related recommendations'. Then, we constructed the PICO of both the related and the reference recommendations (additional details are described in subsection 'Constructing the PICOs'). This allowed us to assess the matching of the PICO of each related recommendation to the PICO corresponding to the reference recommendation (i.e., assess the extent to which the PICOs of the two respective recommendations match). Finally, we assessed the concordance between the recommendations with matching PICOs and explored reasons for any discordance. As part of that exploration, we assessed the methodological quality of the related guidelines.

**Reference guideline and reference recommendations.** The rapid advice guide followed the process outlined in the WHO handbook for guideline development [8] and as described in

'Use of Chest Imaging in the Diagnosis and Management of COVID-19: A WHO Rapid Advice Guide' [5]. The guideline included six questions (two diagnosis questions and four management questions) resulting in seven recommendations. The corresponding recommendations were developed and finalized in May 2020. These recommendations are referred to as reference recommendations.

**Related guidelines and related recommendations.** Eligible guidelines or guidance documents were those issued by guideline developing groups and addressing chest imaging for the screening, diagnosis or management of COVID-19. We excluded documents that were: editorials or opinion pieces; developed by individuals; or dedicated to chest imaging for heart examination, reporting of imaging results, or infection prevention and control measures. We included documents published in any language.

To identify potentially eligible guidelines, we ran a non-systematic search in April 2020, using Google and Google scholar using words such as guideline, guidance, COVID-19, chest imaging, and radiology. We also searched websites of relevant organizations, consulted with content experts and screened the reference lists of the included guidelines.

Two reviewers assessed the eligibility of the guidelines independently and in duplicate. They resolved disagreements through discussion. For each related guideline, we abstracted the following information: name of the guideline, language, guideline developing group, type of group, and country/region.

We included from the related guidelines all 'related recommendations', i.e., formal statements that, on the face of it, matched one of the seven reference recommendations. All challenges with eligibility were resolved with the help of the senior investigator.

**Constructing PICOs.** We constructed the PICOs of the recommendations since the PICOs of the recommendations were not explicitly stated in the guideline and/or the PICOs did not adequately represent the recommendations.

Two reviewers independently constructed the PICO of each of the WHO reference recommendation [5] and their related recommendations, i.e., identified the 'population', 'intervention', 'comparator', 'setting' and 'purpose' elements of the PICO. We considered the first three elements as essential to include in the constructed PICO; we included the last two elements only if included as part of the recommendation. Any disagreement on the construction of the PICO was resolved by discussion, or with the help of the senior investigator. While constructing the PICO, we applied the following rules:

- We considered the comparator as 'no intervention', if it is not specified;

- We considered 'suspected' as an individual with symptoms based on WHO definition [9], unless specified otherwise;

- We considered 'screening' to be for individuals who are asymptomatic, whereas 'diagnosis' to be for individuals with symptoms;

- We distinguished 'screening' (random screening) from targeted screening (screening contacts of patients with suspected or confirmed COVID-19);

- We considered ambiguous elements as non-matching;

- We constructed a distinct PICO for each subgroup included in a recommendation.

**Matching of PICOs.** Two reviewers independently assessed whether the constructed PICO of each related recommendation matched the constructed PICO of the corresponding reference recommendation. Any disagreements on the comparison of these PICOs were

resolved by discussion, or with the help of the senior investigator. This process was applied in two steps: matching of the elements of the PICO, then matching of the whole PICO, as described below.

First, we assessed whether the elements of the related PICO question matched those of the reference PICO question. The elements we considered when matching were the population, intervention, control, setting and purpose. We assessed whether each of the five elements were entirely matching, partly matching, or non-matching:

- 'Entirely matching' meant that the elements were the same, ignoring any differences in terminology used;

- 'Partly matching' meant that the PICO element of the related recommendation was similar to that of the reference recommendation, but either less or more specific (more narrowly or more broadly defined).

- 'Non-matching' meant that the PICO elements did not meet the above requirements.

When the setting and/or purpose were stated in the reference recommendation but not in the related recommendations, we judged it as less specific.

We used color-coding to present the extent of matching as follows: 'green' for entirely matching and 'yellow' for partly matching ('light yellow' for less specific and 'dark yellow' for more specific).

Then we assessed whether the related PICO matched the reference PICO. When at least one of the five elements was 'non-matching', we considered the PICOs as non-matching. When all elements were 'entirely matching', we considered the PICOs as 'entirely matching'. In all other cases, we considered the PICOs as 'partly matching'. We refer to entirely matching or partly matching PICOs as 'matching'.

**Assessing concordance of recommendations.** For this step, we only considered recommendations for which the constructed PICO 'entirely matched' or 'partly matched' the PICO of the reference recommendation. We compared each recommendation to its reference recommendation with regard to strength and direction to assess concordance. We inferred the strength of recommendation of the related recommendation based on the terms used. We interpreted the following terms to imply a strong recommendation: recommend, indicate, should, and must; and the following ones to imply a conditional recommendation: envisage, consider, advised, could, and may. In addition, we examined other terms used in the guidelines to better interpret their use of terminology.

We attempted to assess the reasons for discordance in PICO-matching (partly or entirely matching) recommendations following the methodological approach in Fig 1. In partly PICO-matching recommendations, we explored whether the discordance was explained by reasons related to PICO elements or by reasons related to evidence-to-decision (EtD) factors. In entirely PICO-matching recommendations, we focused on reasons related to evidence-to-decision (EtD) factors only, given the PICO elements do not differ. See Table 1 for more details about those reasons.

**Assessing the methodological quality of guidelines.** We used the Appraisal of Guidelines, Research and Evaluation (AGREE) II instrument to assess the methodological quality of related guidelines with at least one related recommendation [10]. Two reviewers (SY, FJ) applied the AGREE II instrument to each of those guidelines in duplicate and independently. Then, they compared their results and resolved discrepancies when there was a difference of 3 or more points.

**Table 1. Reasons for explaining discordance between related recommendations and reference recommendations.**

| |
|---|
| **Reasons related to EtD factors**[ab] |
| • Different EtD factors considered when developing the recommendation |
| • Different evidence used for the same EtD factor when developing the recommendation |
| • Different judgment made based on the same evidence for the same EtD factor when developing the recommendation |
| **Reasons related to PICO**[c] |
| • At least one of the PICO elements (population, intervention, control, setting or purpose) is narrower in the related recommendation compared to the reference recommendation; a subgroup effect related to this narrower element would justify a different recommendation (e.g., stronger recommendation for a narrowly defined population in the related recommendation compared with a conditional recommendation in a more broadly defined population in the reference recommendation); |
| • At least one of the PICO elements is broader in the related recommendation compared to the reference recommendation; a subgroup effect related to this broader element would justify a different recommendation. |

[a] EtD factors include desirable effects, undesirable effects, values and preferences, certainty of evidence, balance of desirable and undesirable effects, resource use, equity, acceptability, and feasibility.

[b] Applies to both 'entirely PICO-matching recommendations' and 'partly PICO-matching recommendations', the latter defined as: at least one of the PICO elements of the related recommendation is either narrower or broader (i.e., more specific or less specific) than the corresponding PICO element of the reference recommendation.

[c] Applies only to 'partly PICO-matching' recommendations, where that at least one of the PICO elements of the related recommendation is either narrower or broader than the corresponding PICO element of the reference recommendation

## Synthesis

We summarized data related to matching and concordance assessment in both narrative and tabular formats. We reported percentages for categorical variables and medians with ranges for continuous variables. For the methodological quality of guidelines, we calculated the scaled domain scores according to the AGREE II instrument manual, where scores range from 0 to 100%. The six domains of the instrument are as follows: scope and purpose, stakeholder involvement, rigour of development, clarity of presentation, applicability and editorial independence.

## Results

We identified 73 documents potentially eligible as 'related guidelines'. After removing duplicates (n = 5), we excluded 36 documents for the following reasons: represent editorials or opinion statements (n = 19), developed by individuals (n = 2), or do not address using chest imaging for screening, diagnosis or management of COVID-19 (n = 15). We included 32 related guidelines issued by 26 guideline developing groups. A flowchart of the included guidelines is presented in Fig 2. S1 Table describes the characteristics of these 32 guidelines: 88% were developed by professional societies and 63% were in English language.

### Construction of PICOs and PICO matching

From the 32 related guidelines, 22 had at least one related recommendation, with a total of 89 related recommendations. S2 Table shows the results of construction of PICOs and matching of related recommendations for each of the reference recommendations. Table 2 shows the distribution of both related recommendations and PICO-matching recommendations from these 22 guidelines. Out of the 89 related recommendations, 43 partly PICO-matched and only one entirely PICO-matched one of the reference recommendations. We did not identify any PICO-matching recommendation for reference recommendation 6.

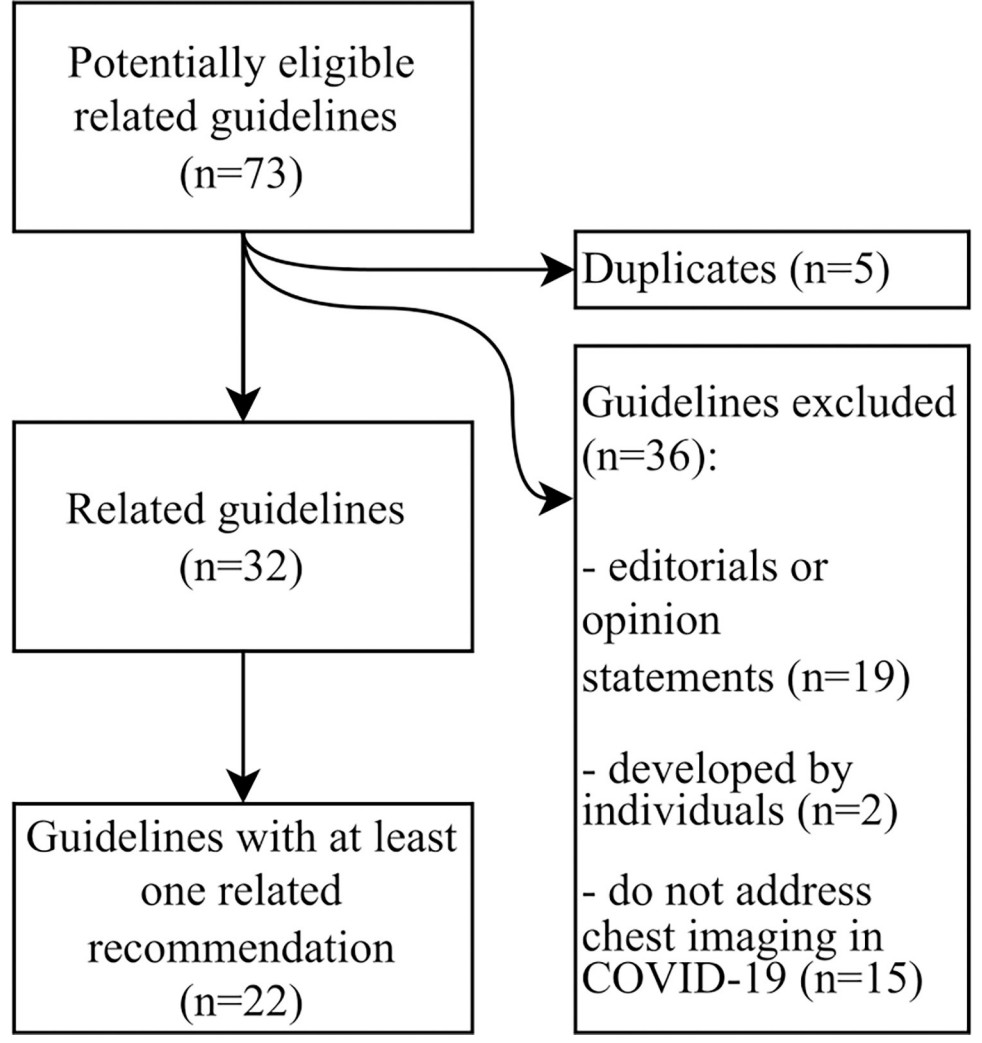

**Fig 2. Flowchart of the included guidelines.**

Reference recommendation 1 is the only recommendation with entirely and partly PICO-matched related recommendations. The entirely PICO-matched recommendation is that of guideline #26 where the elements are the same as the reference's (P: patients with suspected COVID-19 infection, and are asymptomatic, I: imaging, C: no imaging). On the other hand, a partly PICO-matched recommendation is that of guideline #9, where the elements are either broader (P: patients with mild or no symptoms) or narrower (I: CT scan, C: no CT scan) than the reference's.

The seven reference recommendations had a median of 12 related recommendations (range 3–20), a median of 7 PICO-matching recommendations (range 0–13) (Table 3). The 22 guidelines included a median of 4 related recommendations (range 1–7), and a median of 2 PICO-matching recommendations (range 0–6).

## Concordance between recommendations

Out of the 44 PICO-matching recommendations, 8 (18%) were concordant with their reference recommendations (same strength, same direction). Of the 36 discordant

**Table 2. Map summarizing the matching of the related recommendations to each of the 7 reference recommendations.**

| Related guideline | Reference recommendations | | | | | | | Related recommendations | PICO-matching recommendations |
|---|---|---|---|---|---|---|---|---|---|
| | R1 | R2.1 | R2.2 | R3 | R4 | R5 | R6 | | |
| [G1] | ✓[a] | ✓ | ✓ | ✓ | ✓ | X | X | 6 | 4 |
| [G2] | X | X | X | ✓ | X | X | X | 1 | 0 |
| [G3] | ✓ | ✓ | ✓ | X | X | ✓ | X | 4 | 2 |
| [G5] | X | ✓ | ✓ | X | X | X | X | 2 | 2 |
| [G8] | X | X | X | X | X | ✓ | X | 1 | 0 |
| [G9] | ✓ | ✓ | ✓ | X | X | X | ✓ | 4 | 3 |
| [G10] | X | X | X | ✓ | ✓ | X | X | 2 | 0 |
| [G12] | ✓ | ✓ | ✓ | ✓ | X | ✓ | X | 5 | 3 |
| [G13] | ✓ | ✓ | ✓ | ✓ | X | X | X | 4 | 1 |
| [G14] | X | X | X | X | X | ✓ | X | 1 | 0 |
| [G17] | ✓ | ✓ | ✓ | X | ✓ | X | ✓ | 5 | 2 |
| [G18] | ✓ | ✓ | ✓ | ✓[b] | ✓ | ✓ | X | 7 | 6 |
| [G19] | ✓[a] | ✓ | ✓ | X | ✓ | X | X | 4 | 1 |
| [G21] | ✓ | ✓ | ✓ | ✓ | ✓ | X | X | 5 | 2 |
| [G22] | X | X | X | ✓ | ✓ | X | X | 2 | 0 |
| [G24] | ✓ | ✓ | ✓ | X | X | ✓ | ✓ | 5 | 3 |
| [G25] | X | ✓ | ✓ | X | X | X | X | 2 | 2 |
| [G26] | ✓ | ✓ | ✓ | ✓ | ✓ | ✓ | X | 6 | 4 |
| [G28] | ✓ | ✓ | ✓ | X | X | ✓ | X | 4 | 2 |
| [G29] | ✓ | ✓ | ✓ | X | ✓ | ✓ | X | 5 | 0 |
| [G30] | X | ✓[b] | ✓[b] | ✓ | ✓ | ✓ | X | 7 | 4 |
| [G32] | ✓ | ✓ | ✓[c] | ✓ | ✓ | X | X | 7 | 3 |

Note: '✓' indicates the presence of at least one related recommendation, while 'X' indicates the absence of a related recommendation. Green indicates entirely matching PICOs, yellow indicates partly matching PICOs, red indicates non-matching PICOs. References of related guidelines are included in the supplemental file.

[a] Includes two related recommendations, with only one partly PICO-matching recommendation

[b] Includes two related recommendations with two partly PICO-matching recommendations

[c] Includes three related recommendations

recommendation, 28 (78%) had the same direction but a different strength, and 8 (22%) had a different direction. The full detailed are included in S3 Table. The strength of the recommendation was explicitly reported in only one recommendation and was not consistent with the reference recommendation.

**Table 3. Summary of the matching and concordance of the related recommendations to each of the seven reference recommendations.**

| | Related recommendations | Non-matching recommendations | Partly PICO-matching recommendations | Entirely PICO-matching recommendations | Concordant recommendations |
|---|---|---|---|---|---|
| R1 | 15 | 8 | 6 | 1 | 4 |
| R2.1 | 18 | 7 | 11 | 0 | 1 |
| R2.2 | 20 | 7 | 13 | 0 | 2 |
| R3 | 12 | 5 | 7 | 0 | 0 |
| R4 | 11 | 8 | 3 | 0 | 0 |
| R5 | 10 | 7 | 3 | 0 | 1 |
| R6 | 3 | 3 | 0 | 0 | 0 |

The seven reference recommendations had a median of 1 concordant recommendation (range 0–4), while the 22 guidelines included a median of 0 concordant recommendations (range 0–4).

We attempted to assess the reasons for discordance between the recommendations, however the reference recommendations did not report the needed information to assess the EtD factors in the related recommendations.

## Methodological quality of guidelines

Table 4 shows the results of the methodological quality assessment for each of the six domains of the AGREE II instrument, for the 22 related guidelines with at least one related recommendation. The domain with the lowest scaled score was #6 'editorial independence' with a median of 0% (range 0–63). The domain with the highest scaled score was #4 'clarity of presentation' with a median of 63% (range 19–100). While one guideline had an overall score of 72%, all other guidelines had overall scores of 47% or less, and the median of the overall scores was 21% (range 6–72). The limited variability in the distribution of scores did not allow us to explore whether concordance was associated with methodological quality of the guidelines.

## Discussion

The objective of this study was to describe a methodological approach to explore the concordance of recommendations across guidelines and its application to the case of the WHO

**Table 4. The scores of the methodological quality assessment of the related guidelines using the AGREE II instrument.**

| Guideline | Domain 1. Scope and Purpose | Domain 2. Stakeholder involvement | Domain 3. Rigour of development | Domain 4. Clarity of presentation | Domain 5. Applicability | Domain 6. Editorial independence | Overall score[a] |
|---|---|---|---|---|---|---|---|
| [G1] | 64 | 11 | 4 | 64 | 19 | 38 | 33 |
| [G2] | 6 | 3 | 1 | 36 | 2 | 0 | 8 |
| [G3] | 6 | 3 | 6 | 22 | 4 | 0 | 7 |
| [G5] | 8 | 11 | 1 | 44 | 2 | 0 | 11 |
| [G8] | 100 | 89 | 68 | 100 | 13 | 63 | 72 |
| [G9] | 64 | 39 | 3 | 42 | 2 | 0 | 25 |
| [G10] | 33 | 28 | 10 | 81 | 0 | 0 | 25 |
| [G12] | 22 | 3 | 1 | 36 | 0 | 0 | 10 |
| [G13] | 39 | 3 | 1 | 67 | 2 | 0 | 19 |
| [G14] | 3 | 14 | 5 | 61 | 4 | 0 | 15 |
| [G17] | 33 | 56 | 3 | 64 | 2 | 0 | 26 |
| [G18] | 67 | 33 | 17 | 83 | 4 | 0 | 34 |
| [G19] | 22 | 28 | 2 | 33 | 13 | 0 | 16 |
| [G21] | 44 | 50 | 23 | 100 | 2 | 0 | 37 |
| [G22] | 25 | 39 | 8 | 67 | 0 | 0 | 23 |
| [G24] | 31 | 14 | 1 | 44 | 4 | 0 | 16 |
| [G25] | 8 | 3 | 3 | 19 | 0 | 0 | 6 |
| [G26] | 0 | 11 | 2 | 47 | 0 | 0 | 10 |
| [G28] | 22 | 25 | 4 | 64 | 0 | 0 | 19 |
| [G29] | 47 | 28 | 1 | 58 | 2 | 0 | 23 |
| [G30] | 69 | 36 | 13 | 86 | 2 | 0 | 34 |
| [G32] | 89 | 58 | 31 | 94 | 10 | 0 | 47 |
| **Median** | 32 | 26 | 4 | 63 | 2 | 0 | 21 |
| **Range** | [0, 100] | [3, 89] | [1, 68] | [19, 100] | [0, 19] | [0, 63] | [6, 72] |

[a]Mean of the scores of the six domains

recommendations on chest imaging for the diagnosis and management of COVID-19. We identified a total of 89 related recommendations from 22 related guidelines. Following a detailed methodological approach, we found that out of the 89 related recommendations, 43 partly matched and only one entirely matched one of the reference recommendations. Out of these 44 recommendations, 8 were concordant with one of the reference recommendations. When considering the seven reference recommendations, they had a median of 12 related recommendations (range 3–20), a median of 7 PICO-matching recommendations (range 0–13), and a median of 1 concordant recommendation (range 0–4). We were not able to assess the reasons for discordance due to the lack of information on the EtD factors.

This is the first study exploring the concordance between COVID-19 recommendations across guidelines. A strength of this study is the use of a detailed methodological approach that addressed a number of challenges, such as the lack of clear PICO questions. While the approach has face and content validity, we believe it requires further testing and validation, especially that the 'case study' nature of this project did not allow us to test the approach on a larger scale. In addition, the poor reporting of the related guidelines did not allow us to fully test the approach. Specifically, these guidelines did not explicitly report on the strength of the recommendations, and we had to infer that strength based on the terminology used. Also, we were not able to explore the reasons for discordance because of the poor reporting of the factors that drove the strength and direction of the recommendations.

One potential reason for discordance is the poor methodological quality of the guidelines, as evidence by their scoring on the AGREE II instrument. Unfortunately, the limited variability in the distribution of scores did not allow us to test this hypothesis. Other studies that assessed the quality of COVID-19 guidelines have also reported low scores [3, 11–13]. For example, one study found that COVID-19 practice guidelines suffered from poor methodological quality (mean AGREE II overall score was 30%) as well as poor reporting quality (mean overall reporting rates for all was 33%) [11]. It is worth noting that the reference guideline in this study received the highest overall scores in methodological (72.8%) and reporting qualities (83.8%) [11].

Alper et al. systematically assessed the 'consistency' of recommendations across eight guidelines on hypertension management published as of April 2018 [6]. The authors concluded that there was a 'high' rate of inconsistency: out of 68 recommendations, 32% were consistent in direction and strength, 27% were consistent in direction but not in strength, and 41% were inconsistent in direction. Differently from our study, the investigators did not have a 'reference guideline' or 'reference recommendations'. As a result, they had to create a 'reference standard' to compare recommendations against. For that, they used an approach similar to constructing the PICOs, using the 'PIC specifications.' However, they did not assess the extent to which the PICO elements matched between recommendations. Also, they did not explore the reasons for 'inconsistency' between recommendations. A more recent study used the same methodology as Alper et al. to assess the concordance hypertension recommendations in Southeast Asia [7]. They found a high rate of non-concordance with internationally reputable CPGs.

Guideline users seeking guidance on the use of chest imaging for the diagnosis and management of COVID-19 will deal with discordant recommendations. Some of this discordance might be explained by non-matching PICO elements underlying the recommendations. In partly matching recommendations, at least one of the PICO elements of the matching recommendation is either less or more specific than the corresponding PICO elements of the reference recommendations. Unfortunately, these elements are typically not explicitly reported. Another potential explanation is the EtD factors as different factors may be considered, different evidence may be used for the same factor, or different judgment is made based on the same evidence for the same factor. It is also possible that discordance is not explained by either

PICO elements or EtD factors and may be due to differences in processes of guideline development or 'hidden factors' such as conflicts of interest [14].

Our study, particularly the proposed methodological approach, can help others in exploring concordance of recommendations across guidelines. In cases where no 'reference recommendation' exist, one of the early steps in the process would need to be the selection or the development of such a reference recommendation. Guideline groups need to be more explicit about the factors driving their recommendations e.g., by using an evidence-to-decision approach.

Future research should further test the proposed methodological approach and refine it as needed. Also, it would be interesting to assess how actual practice compares to the WHO recommendations, taking advantage of a survey of radiology departments regarding their current practices in the management of patients with COVID-19 that was conducted approximately at the same time the WHO recommendations were being developed [15].

## Supporting information

**S1 Table. Characteristics of related guidelines.**
(DOCX)

**S2 Table. Results of the construction of the PICOs and matching of related recommendations for each of the reference recommendations.** Blue: reference recommendation, Green: entirely matching, Light yellow: partly matching–less specific, and Dark yellow: partly matching–more specific.
(DOCX)

**S3 Table. Concordance of the matching recommendations and the reference recommendations in terms of direction and strength.** Grey: reference recommendation, Red: strongly against, Red: conditional against, Light green: conditional for, and Dark green: strongly for.
(DOCX)

## Acknowledgments

We thank Dr. Maria del Rosario Perez and Dr. Emilie van Deventer from the World Health Organization for their support throughout the study.

## Author Contributions

**Conceptualization:** Sally Yaacoub, Fatimah Chamseddine, Ivana Blazic, Guy Frija, Elie A. Akl.

**Data curation:** Sally Yaacoub, Fatimah Chamseddine.

**Formal analysis:** Sally Yaacoub.

**Investigation:** Sally Yaacoub, Fatimah Chamseddine, Farah Jaber.

**Methodology:** Sally Yaacoub, Fatimah Chamseddine, Elie A. Akl.

**Supervision:** Elie A. Akl.

**Validation:** Elie A. Akl.

**Visualization:** Sally Yaacoub, Elie A. Akl.

**Writing – original draft:** Sally Yaacoub, Elie A. Akl.

**Writing – review & editing:** Fatimah Chamseddine, Farah Jaber, Ivana Blazic, Guy Frija, Elie A. Akl.

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
