## [Decision Letter · Decision Letter 0]

19 Dec 2022

PONE-D-22-31128Exploring the concordance of recommendations across guidelines: a proposed methodological approach and a case studyPLOS ONE

Dear Dr. Akl,

Thank you for submitting your manuscript to PLOS ONE. After careful consideration, we feel that it has merit but does not fully meet PLOS ONE’s publication criteria as it currently stands. Therefore, we invite you to submit a revised version of the manuscript that addresses the points raised during the review process.

We look forward to receiving your revised manuscript.

Kind regards,

Erik Loeffen, M.D., Ph.D.

Academic Editor

PLOS ONE

Journal Requirements:

Additional Editor Comments:

Thank you for this study, it is interesting howevere it does need some work to be suitable for publication in PLOS One. Specifically, pay close detail to the suggestions made by the reviewers. In addition: 1) what were your methods based upon? existing studies? group consensus? how will you know if these are solid? 2) please omit the term PICOstructed, it implies that it is an established concept, however it is just constructing a PICO, please call it as it is, 3) why is in Table 2 green and light-green used while the texts writes about green and yellow (for partly)?, 4) although it is touched upon in the discussion, I find it a missed opportunity that the quality of the guidelines is not taken into acocunt when interpreting the concordances. It would be valuable to know if the higher quality guidelines shower greater concordance. Especially since most guidelines have already been GRADEd (as authors describe) which might be suitable to use for this study. I would suggest authors to reconsider incorporating this, although it would not be a go/no-go for publication, it would make the results way more valuable for the scientific community as a whole.

Good luck, I would be happy to see the revised version.

Reviewers' comments:

Reviewer's Responses to Questions

**Comments to the Author**

1. Is the manuscript technically sound, and do the data support the conclusions?

Reviewer #1: No

Reviewer #2: Yes

2. Has the statistical analysis been performed appropriately and rigorously? 

Reviewer #1: N/A

Reviewer #2: Yes

3. Have the authors made all data underlying the findings in their manuscript fully available?

Reviewer #1: Yes

Reviewer #2: Yes

4. Is the manuscript presented in an intelligible fashion and written in standard English?

Reviewer #1: Yes

Reviewer #2: Yes

5. Review Comments to the Author

Reviewer #1: Thank you for inviting me to review this manuscript exploring an approach to examine concordance between guideline recommendations for specific clinical scenarios. This is an interesting topic; incongruity between recommendations in guidelines is commonly cited as a barrier to guideline uptake in clinical practice and exploring this aspect of guideline recommendations is important. I do have some comments for the authors to consider that can be broadly categorized as position of the study and generalizability, and clarity around methods.

Position and generalizability of this study. The authors aimed to develop a methodological approach to examine concordance between guideline recommendations for a given clinical issue and applying it to a case study (chest imaging for COVID-19 patients) but it seems more like the reverse; examining concordance between recommendations for chest imaging for patients with COVID-19 that may be generalizable to other clinical scenarios. Could the authors explore in greater detail how this can be generalized to other clinical scenarios to position the study as a new methodology rather than a case study?

With regards to generalizability, the authors note that guidelines developed during the COVID-19 pandemic were different than other guidelines due to the rapidly evolving situation and the sparsity of evidence (high quality or otherwise) due to the novel SARS virus (COVID-19). Could you explore how this approach could be applied to guidelines outside of the context of COVID-19 guidelines?

Finally, could the authors explore the utility of this methodological approach for managing patients using guideline recommended care? What are guideline users to do if they find, using the proposed methodology, that recommendations between guidelines are not concordant? What recommendations are they to follow and how is it to guide clinical care (or improve care)?

Methods. What definition of a guideline did the authors use? There are different definitions and terminology used for guidelines (e.g., guideline, pathway, position statement) and knowing the operational definition used by the authors for identifying guidelines to include would be helpful for the reader. Similarly, did the authors consider assessing the quality of the included guidelines using the AGREEII tool? Understanding the quality of the guidelines may provide additional insight into concordance and reasons for incongruence in recommendations, which was noted by the authors as a limitation of the current study. Collectively, these two issues speak to a concern about comparability of the included guidelines in terms of development and scope; can the authors speak to the comparability of the guidelines across the AGREEII domains to help the reader understand how comparable the guidelines were with regards to developing the recommendations?

Could the authors describe their search strategy in greater detail to help the reader understand their approach to identifying potentially eligible guidelines? Similarly, what were the eligibility criteria?

Minor comments:

Title – could be more informative regarding the specific guidelines used/ recommendations.

What was the rationale behind choosing this topic as a case study?

Some minor grammatical errors throughout.

Reviewer #2: Thank you for submitting this article to the PLOS ONE. I was pleased to receive it as a reviewer.

I have the following questions for you, which I believe, need to be addressed before publication:

The statistical analysis should be reported according to the recently published guidelines:

Blackstone EH and Weisel RD. The conclusion of papers published in the Journal should be supported by an appropriate statistical analysis. J Thorac Cardiovasc Surg. 2014;148:2479.

Huebner M, Vach W, le Cessie S. A systematic approach to initial data analysis is good research practice. J Thorac Cardiovasc Surg. 2016;151:25-7.

Wasserstein RL, Lazar NA. The ASA's Statement on p-Values: Context, Process, and Purpose. The American Statistician. 2016;70:2, 129-133.

Greenland S, Senn SJ, Rothman KJ, et. al. Statistical Tests, P-values, Confidence Intervals, and Power: A Guide to Misinterpretations. The American Statistician. 2016;70:2. Suppl 1:1-12.

McMurry TL, Hu Y, Blackstone EH, Kozower BD. Propensity scores: Methods, considerations, and applications in the Journal of Thoracic and Cardiovascular Surgery. J Thorac Cardiovasc Surg. 2015;150:14-9

Winger DG, Nason KS. Propensity-score analysis in thoracic surgery: When, why, and an introduction to how. J Thorac Cardiovasc Surg. 2016;151:1484-7.

Bagiella E. Use (and misuse) of instrumental variables. J Thorac Cardiovasc Surg. 2015;150:460.

Bagiella E, Karamlou T, Chang H, Spivack J. Instrumental variable methods in clinical research. J Thorac Cardiovasc Surg. 2015;150:779-82.

Rajeswaran J, Blackstone EH. Patient-reported outcomes and importance of their appropriate statistical analyses. J Thorac Cardiovasc Surg. 2015;150:461-2.

There are typo errors in the text. Please thoroughly check the article.

The manuscript should be reported according to the information to authors.

Good luck with your article, and thanks again for submitting it.

6. PLOS authors have the option to publish the peer review history of their article (what does this mean?). If published, this will include your full peer review and any attached files.

Reviewer #1: **Yes: **Khara Sauro

Reviewer #2: No

---

## [Author Response · Author response to Decision Letter 0]

2 Feb 2023

Dear Dr. Chenette and Dr. Loeffen,

We thank you for the opportunity to revise our manuscript for your consideration for publication in the PLOS One. The comments and suggestions of the Academic Editor and reviewers have helped us to improve the quality of our manuscript. Please note that to address the Academic Editor and first Reviewer’s request to critically appraise the quality of the included guidelines, we had to recruit Dr. Farah Jaber to help with that task which we did in duplicate. Based on her contribution, Dr. Jaber meets the ICMJE criteria for authorship. 

Please find on the following pages our detailed point-by-point responses to the suggestions. We will be happy to address any further comments or suggestions you or the reviewers might have.

With kind regards,

Elie A. Akl, MD, MPH, PhD

Department of Internal Medicine

American University of Beirut 

P.O.Box 11-0236 / CRI (E15)

Riad-El-Solh Beirut 1107 2020

Beirut – Lebanon

T: + 961 1 374374

Email: ea32@aub.edu.lb

The Academic Editor’s and Reviewers’ comments are in bold font and our replies in regular font. Extracts from the text are in italic fonts with changes underlined. We have indicated the sections where revisions have been made in our manuscript.

Academic Editor: 

Thank you for this study, it is interesting however it does need some work to be suitable for publication in PLOS One. Specifically, pay close detail to the suggestions made by the reviewers. 

Response: Thank you for the opportunity. We have carefully addressed all the suggestions made by the reviewers.

In addition: 

1) What were your methods based upon? existing studies? group consensus? how will you know if these are solid?

Response: We based our methodological approach based on review of the existing literature and on group consensus achieved through discussions amongst the authors. We added the following in the Methods section: 

“We drafted the methodological approach based on a review of the literature, discussion amongst the authors and iterative revisions when applying it to the WHO recommendations on chest imaging for the diagnosis and management of COVID-19.”

While the approach has face and content validity, we believe it requires further testing and validation. We have clarified this point in the discussion section.

While the approach has face and content validity, we believe it requires further testing and validation, especially that the ‘case study’ nature of this project did not allow us to test the approach on a larger scale.

2) Please omit the term PICOstructed, it implies that it is an established concept, however it is just constructing a PICO, please call it as it is, 

Response: We have omitted the term from the paper and now say ‘constructed a PICO’.

3) Why is in Table 2 green and light-green used while the texts writes about green and yellow (for partly)? 

Response: Thank you for noting this. We have modified the colors of table 2 to be consistent with the text.

“Green indicates entirely matching PICOs, yellow indicates partly matching PICOs, red indicates non-matching PICOs.”

4) Although it is touched upon in the discussion, I find it a missed opportunity that the quality of the guidelines is not taken into acocunt when interpreting the concordances. It would be valuable to know if the higher quality guidelines shower greater concordance. Especially since most guidelines have already been GRADEd (as authors describe) which might be suitable to use for this study. I would suggest authors to reconsider incorporating this, although it would not be a go/no-go for publication, it would make the results way more valuable for the scientific community as a whole.

Response: We appreciate this point raised by the academic editor and reviewer #1. Accordingly, we have now critically appraised the related guidelines using the Appraisal of Guidelines for Research & Evaluation (AGREE) II instrument. 

Subsequently, we added the following text to the methodological approach subsection: 

“As part of that exploration, we assessed the methodological quality of the related guidelines.” 

We added at the end of the methods section a subsection titled ‘Assessing the methodological quality of guidelines’:

“We used the Appraisal of Guidelines, Research and Evaluation (AGREE) II instrument to assess the methodological quality of related guidelines with at least one related recommendation. Two reviewers (SY, FJ) applied the AGREE II instrument to each of those guidelines in duplicate and independently. Then they compared their results and resolved discrepancies when there was a difference of 3 or more points.” 

We also added to the subsection ‘Synthesis’ the following:

“The scaled domain scores were calculated according to the AGREE II instrument manual, where scores range from 0 to 100%. The six domains of the instrument are as follows: scope and purpose, stakeholder involvement, rigour of development, clarity of presentation, applicability and editorial independence.”

 We added to the results section a subsection titled: “Methodological quality of guidelines”

“Table 4 shows the results of the methodological quality assessment for each of the six domains of the AGREE II instrument, for the 22 related guidelines with at least one related recommendation. The domain with the lowest scaled score was #6 ‘editorial independence’ with a median of 0% (range 0-63). The domain with the highest scaled score was #4 ‘clarity of presentation’ with a median of 63% (range 19-100). While one guideline had an overall score of 72%, all other guidelines had overall scores of 47% or less, and the median of the overall scores was 21% (range 6-72). The limited variability in the distribution of scores did not allow us to explore whether concordance was associated with methodological quality of the guidelines.”

We added the following text to the discussion section:

“One potential reason for discordance is the poor methodological quality of the guidelines, as evidence by their scoring on the AGREE II instrument. Unfortunately, the limited variability in the distribution of scores did not allow us to test this hypothesis. Other studies that assessed the quality of COVID-19 guidelines have also reported low scores [3,8-10]”

 

Reviewer #1:

Thank you for inviting me to review this manuscript exploring an approach to examine concordance between guideline recommendations for specific clinical scenarios. This is an interesting topic; incongruity between recommendations in guidelines is commonly cited as a barrier to guideline uptake in clinical practice and exploring this aspect of guideline recommendations is important. I do have some comments for the authors to consider that can be broadly categorized as position of the study and generalizability, and clarity around methods.

Response: Thank you for the positive evaluation and interest.

Position and generalizability of this study:

The authors aimed to develop a methodological approach to examine concordance between guideline recommendations for a given clinical issue and applying it to a case study (chest imaging for COVID-19 patients) but it seems more like the reverse; examining concordance between recommendations for chest imaging for patients with COVID-19 that may be generalizable to other clinical scenarios. Could the authors explore in greater detail how this can be generalized to other clinical scenarios to position the study as a new methodology rather than a case study?

Response: Thank you for raising this point. It is a relevant and useful perspective that we have reflected by adding the following text to the beginning of the methods section:

“We drafted the methodological approach based on a review of the literature, discussion amongst the authors and iterative revisions when applying it to the WHO recommendations on chest imaging for the diagnosis and management of COVID-19.”

In line with that, we have slightly modified the title to say: 

“a proposed methodological approach based on a case study”

We have also modified the objective to say:

“To describe a methodological approach to explore the concordance of recommendations across guidelines and its application to the case of the WHO recommendations on chest imaging for the diagnosis and management of COVID-19.”

With regards to generalizability, the authors note that guidelines developed during the COVID-19 pandemic were different than other guidelines due to the rapidly evolving situation and the sparsity of evidence (high quality or otherwise) due to the novel SARS virus (COVID-19). Could you explore how this approach could be applied to guidelines outside of the context of COVID-19 guidelines?

Response: Thank you for the comment. The methodological approach was developed and applied to COVID-19 guidelines. However, we believe that it can also be applied in different clinical contexts, given that the steps are not specific to COVID-19, and are applicable to any health topic (i.e., identifying reference and related guidelines, identifying reference and related recommendations, constructing the PICOs, matching of PICOs, assessing the concordance of recommendations and possible reasons for discordance). Methodologies with related aims (e.g. the one by Alper et al. that we address in the discussion), have used similar steps to ours and been applied to different topics (e.g., treatment of hypertension). At the same time, and as mentioned earlier, we acknowledge that the approach requires further testing in different contexts. 

Finally, could the authors explore the utility of this methodological approach for managing patients using guideline recommended care? What are guideline users to do if they find, using the proposed methodology, that recommendations between guidelines are not concordant? What recommendations are they to follow and how is it to guide clinical care (or improve care)?

Response: Thank you for raising this interesting point. The primary objective of this approach is to assess concordance between recommendations but it is not specifically for users to choose which of the recommendations to follow in the clinical care context. For the latter objective, users can consider a number of factors, such as the relevance of the recommendations to the clinical questions, the quality of the guidelines, and their up-to-date-ness of the guidelines. 

Methods. What definition of a guideline did the authors use? There are different definitions and terminology used for guidelines (e.g., guideline, pathway, position statement) and knowing the operational definition used by the authors for identifying guidelines to include would be helpful for the reader. 

Response: The reviewer raises an important point. We did not restrict our definition of ‘guideline’ to that defined in the literature (e.g., Institute of Medicine), given that our study was conducted in the early stages of the pandemic when guidance documents of various definitions were published. However, we considered documents issued by guideline developing groups (rather than by individuals) that provide formal statements on the topic in question (i.e., chest imaging for the screening, diagnosis or management of COVID-19). 

Similarly, did the authors consider assessing the quality of the included guidelines using the AGREEII tool? Understanding the quality of the guidelines may provide additional insight into concordance and reasons for incongruence in recommendations, which was noted by the authors as a limitation of the current study. Collectively, these two issues speak to a concern about comparability of the included guidelines in terms of development and scope; can the authors speak to the comparability of the guidelines across the AGREEII domains to help the reader understand how comparable the guidelines were with regards to developing the recommendations?

Response: We appreciate this point raised by the academic editor and reviewer #1. Accordingly, we have now critically appraised the related guidelines using the Appraisal of Guidelines for Research & Evaluation (AGREE) II instrument. 

Subsequently, we added the following text to the methodological approach subsection: 

“As part of that exploration, we assessed the methodological quality of the related guidelines” 

We added at the end of the methods section a subsection titled ‘Assessing the methodological quality of guidelines’:

“We used the Appraisal of Guidelines, Research and Evaluation (AGREE) II instrument to assess the methodological quality of related guidelines with at least one related recommendation. Two reviewers (SY, FJ) applied the AGREE II instrument to each of those guidelines in duplicate and independently. Then they compared their results and resolved discrepancies when there was a difference of 3 or more points. The scaled domain scores were calculated according to the AGREE II instrument manual, where scores range from 0 to 100%. The six domains of the instrument are as follows: scope and purpose, stakeholder involvement, rigour of development, clarity of presentation, applicability and editorial independence.”

 We added to the results section a subsection titled: “Methodological quality of guidelines”

“Table 4 shows the results of the methodological quality assessment for each of the six domains of the AGREE II instrument, for the 22 related guidelines with at least one related recommendation. The domain with the lowest scaled score was #6 ‘editorial independence’ with a median of 0% (range 0-63). The domain with the highest scaled score was #4 ‘clarity of presentation’ with a median of 63% (range 19-100). While one guideline had an overall score of 72%, all other guidelines had overall scores of 47% or less, and the median of the overall scores was 21% (range 6-72). The limited variability in the distribution of scores did not allow us to explore whether concordance was associated with methodological quality of the guidelines.”

We added the following text to the discussion section:

“One potential reason for discordance is the poor methodological quality of the guidelines, as evidence by their scoring on the AGREE II instrument. Unfortunately, the limited variability in the distribution of scores did not allow us to test this hypothesis. Other studies that assessed the quality of COVID-19 guidelines have also reported low scores [3,8-10]”

Could the authors describe their search strategy in greater detail to help the reader understand their approach to identifying potentially eligible guidelines? Similarly, what were the eligibility criteria?

Response: Thank you for the comment. We did not conduct a systematic search to identify potentially eligible guidelines. Given that guidelines are generally published on the websites of the guideline-producing organizations specifically in such urgent settings, we searched Google and Google scholar using words such as guideline, guidance, COVID-19, SARS-CoV, chest imaging, radiology, and others. We also searched websites of relevant organizations, consulted with content experts and screened the reference lists of the included guidelines. 

We further clarified our search as follows: 

“To identify potentially eligible guidelines, we ran a non-systematic search in April 2020 using Google and Google scholar using words such as guideline, guidance, COVID-19, SARS-CoV, chest imaging, and radiology. We also searched websites of relevant organizations, consulted with content experts and screened the reference lists of the included guidelines.”

We specify in section ‘Related guidelines and related recommendations’ our eligibility criteria. The inclusion criteria were: guidelines or guidance documents issued by guideline developing groups and addressing chest imaging for the screening, diagnosis or management of COVID-19. Whereas, the exclusion criteria were: editorials or opinion pieces; developed by individuals; or dedicated to chest imaging for heart examination, reporting of imaging results, or infection prevention and control measures. 

Minor comments:

Title – could be more informative regarding the specific guidelines used/ recommendations.

Response: Thank you for the suggestion. We have changed the title into the following:

“Exploring the concordance of recommendations across guidelines on chest imaging for the diagnosis and management of COVID-19: a proposed methodological approach based on a case study”

What was the rationale behind choosing this topic as a case study?

Response: The rationale behind choosing this topic was that the group of authors was in charge of coordinating the development of the WHO guideline on the use of chest imaging for the diagnosis and management of COVID-19, where preparatory work for the guideline development included searching for related guidance. After completion, we went back to compare our recommendations to the previously identified related recommendations which entailed the development of a methodological approach. We clarify these points in the Introduction section.

Some minor grammatical errors throughout.

Response: Thank you. We corrected the errors throughout the manuscript.

 

Reviewer #2: 

Thank you for submitting this article to the PLOS ONE. I was pleased to receive it as a reviewer. Good luck with your article, and thanks again for submitting it.

Response: Thank you for the positive feedback.

I have the following questions for you, which I believe, need to be addressed before publication:

The statistical analysis should be reported according to the recently published guidelines:

• Blackstone EH and Weisel RD. The conclusion of papers published in the Journal should be supported by an appropriate statistical analysis. J Thorac Cardiovasc Surg. 2014;148:2479.

• Huebner M, Vach W, le Cessie S. A systematic approach to initial data analysis is good research practice. J Thorac Cardiovasc Surg. 2016;151:25-7.

• Wasserstein RL, Lazar NA. The ASA's Statement on p-Values: Context, Process, and Purpose. The American Statistician. 2016;70:2, 129-133.

• Greenland S, Senn SJ, Rothman KJ, et. al. Statistical Tests, P-values, Confidence Intervals, and Power: A Guide to Misinterpretations. The American Statistician. 2016;70:2. Suppl 1:1-12.

• McMurry TL, Hu Y, Blackstone EH, Kozower BD. Propensity scores: Methods, considerations, and applications in the Journal of Thoracic and Cardiovascular Surgery. J Thorac Cardiovasc Surg. 2015;150:14-9

• Winger DG, Nason KS. Propensity-score analysis in thoracic surgery: When, why, and an introduction to how. J Thorac Cardiovasc Surg. 2016;151:1484-7.

• Bagiella E. Use (and misuse) of instrumental variables. J Thorac Cardiovasc Surg. 2015;150:460.

• Bagiella E, Karamlou T, Chang H, Spivack J. Instrumental variable methods in clinical research. J Thorac Cardiovasc Surg. 2015;150:779-82.

• Rajeswaran J, Blackstone EH. Patient-reported outcomes and importance of their appropriate statistical analyses. J Thorac Cardiovasc Surg. 2015;150:461-2.

Response: Thank you for these references. Our synthesis was limited to summarizing data related to matching and concordance assessment in both narrative and tabular formats. We only reported percentages for categorical variables and medians with ranges for continuous variables. As we did not conduct any statistical tests to determine whether to reject a hypothesis or not, these references are less relevant to our synthesis. We further elaborated on our synthesis as follows:

“We summarized data related to matching and concordance assessment in both narrative and tabular formats. We reported percentages for categorical variables and medians with ranges for continuous variables. For the methodological quality of guidelines, we calculated the scaled domain scores according to the AGREE II instrument manual, where scores range from 0 to 100%. The six domains of the instrument are as follows: scope and purpose, stakeholder involvement, rigour of development, clarity of presentation, applicability and editorial independence.”

There are typo errors in the text. Please thoroughly check the article.

Response: Thank you. We corrected the errors throughout the manuscript.

The manuscript should be reported according to the information to authors.

Reponse: Thank you for raising this point. We have formatted the manuscript according to PLoS One’s requirements.

---

## [Decision Letter · Decision Letter 1]

26 Jun 2023

Exploring the concordance of recommendations across guidelines on chest imaging for the diagnosis and management of COVID-19: a proposed methodological approach based on a case study

PONE-D-22-31128R1

Dear Dr. Akl,

We’re pleased to inform you that your manuscript has been judged scientifically suitable for publication and will be formally accepted for publication once it meets all outstanding technical requirements.

Kind regards,

Hanna Landenmark

Staff Editor

PLOS ONE

Additional Editor Comments (optional):

Reviewers' comments:

Reviewer's Responses to Questions

**Comments to the Author**

1. If the authors have adequately addressed your comments raised in a previous round of review and you feel that this manuscript is now acceptable for publication, you may indicate that here to bypass the “Comments to the Author” section, enter your conflict of interest statement in the “Confidential to Editor” section, and submit your "Accept" recommendation.

Reviewer #2: All comments have been addressed

2. Is the manuscript technically sound, and do the data support the conclusions?

Reviewer #2: Yes

3. Has the statistical analysis been performed appropriately and rigorously? 

Reviewer #2: Yes

4. Have the authors made all data underlying the findings in their manuscript fully available?

Reviewer #2: Yes

5. Is the manuscript presented in an intelligible fashion and written in standard English?

Reviewer #2: Yes

6. Review Comments to the Author

Reviewer #2: (No Response)

7. PLOS authors have the option to publish the peer review history of their article (what does this mean?). If published, this will include your full peer review and any attached files.

Reviewer #2: No

---

## [Editor Report · Acceptance letter]

20 Jul 2023

PONE-D-22-31128R1 

Exploring the concordance of recommendations across guidelines on chest imaging for the diagnosis and management of COVID-19: a proposed methodological approach based on a case study 

Dear Dr. Akl:

I'm pleased to inform you that your manuscript has been deemed suitable for publication in PLOS ONE. Congratulations! Your manuscript is now with our production department. 

Kind regards, 

on behalf of

Dr. Hanna Landenmark 

Staff Editor

PLOS ONE